# SOUNDNESS-AWARE LEVEL: A MICROSCOPIC SIGNATURE THAT PREDICTS LLM REASONING POTENTIAL

## ABSTRACT

Reinforcement learning with verifiable rewards (RLVR) can elicit strong reasoning in large language models (LLMs), while their performance after RLVR varies dramatically across different base models. This raises a fundamental question: what microscopic property of pre-trained models leads to this variation? To investigate, we formalize reasoning as chains of Horn clauses ("if-then" rules) built from features extracted from the LLM's latent space via cross-layer sparse autoencoders (SAEs). We estimate the transition probabilities between its features, and further categorize each rule by its semantic soundness level (e.g., strict, plausible, noisy) with an LLM. Our key discovery is that high-potential models are inherently soundness-aware: their internal probability distributions systematically shift across rules' soundness levels, becoming highly distinct for "strict" versus "noisy" rules. In contrast, weaker models are soundness-agnostic, collapsing to one distribution regardless of soundness levels. To quantify this, we introduce the Soundness-Aware Level (SAL), a microscopic metric using the Jensen-Shannon Divergence to measure the separation between these distributions. We show that the correlation between a model's SAL and its post-RLVR reasoning performance is well described by an exponential power relationship ($R^2 = 0.87$) across families (Qwen, Mistral, Llama, DeepSeek) and scales (0.5B–14B). This reveals that a model's reasoning potential is tied to its intrinsic, pre-trained ability to distinguish sound knowledge from unsound ones. These findings underscore the critical role of model pre-training in shaping reasoning and offer a practical metric grounded in the model's internal mechanisms for selecting / designing stronger base models.

## 1 INTRODUCTION

Large Reasoning Models (LRMs) have markedly shown a strong reasoning capability on mathematical and programming tasks by introducing a specialized "thinking" stage prior to the final answer (Guo et al., 2025; Team, 2025). LRMs are typically trained from general pre-trained large language models (LLMs) via reinforcement learning with verifiable rewards (RLVR). However, empirical studies show that applying the same RLVR pipeline to different pre-trained models can produce substantial disparities in reasoning capabilities (Zeng et al., 2025a). This inconsistency raises a central question: what distinguishes pre-trained models that can be trained into strong LRMs from those that cannot? Pre-training corpora comprise a diverse mix of sound knowledge (e.g., from textbooks) and unsound knowledge (e.g., from low-quality websites). Therefore, we hypothesize that the crucial difference is *microscopic*: a model's intrinsic ability to distinguish this sound knowledge from the unsound. This paper investigates this hypothesis, arguing that an internal, mechanistic perspective offers a path toward a more systematic understanding of what enables complex reasoning.

Prior attempts to explain these disparities have largely focused on macroscopic patterns in the generated texts. Specifically, they analyze behaviors of reasoning, such as the diversity of cognitive phrases (Gandhi et al., 2025; Yue et al., 2025b), the cyclic structure of thought processes (Minegishi et al., 2025), or the model's output uncertainty (Cui et al., 2025; Cheng et al., 2025). While insightful, these approaches measure the downstream effects of reasoning rather than its core mechanism. More recent microscopic analyses have begun to map the internal circuits of reasoning via feature-level case studies (Lindsey et al., 2025a; Ameisen et al., 2025b). However, this line of work has remained primarily qualitative, leaving a crucial gap: a quantitative and scalable method to assess the semantic quality (or soundness) of a model's internal rules and connect it to reasoning potential.

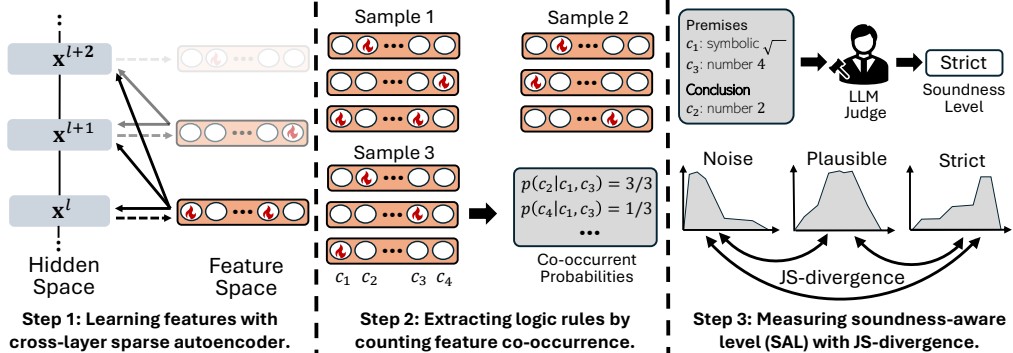

Figure 1: An illustration of our method for probing the internal logic of a pre-trained LLM. The process extracts and analyzes the reasoning rules that a model has already learned from its pre-training. **Step 1:** A cross-layer sparse autoencoder reads out the semantically meaningful features from the LLM's hidden activations. **Step 2:** By tracking feature co-occurrences, we extract the implicit logic rules the model has learned (e.g., $c_1 \wedge c_3 \rightarrow c_2$), estimating the conditional probabilities it has assigned for these entailments. **Step 3:** We then assess the quality of extracted rules. Each rule is labeled with a soundness level by an LLM judge, and we compute the Soundness-Aware Level (SAL) by measuring the JS-divergence between the distributions of different soundness levels. A larger SAL indicates the pre-trained model has more effectively learned to separate its sound knowledge from unsound ones, which in turn predicts its future reasoning potential.

To fill this gap, we propose a new framework to quantify a model's reasoning potential from its internal representations. We first formalize microscopic reasoning as a chain of logic rules, adapting the notion of directional entailment from logic programming, where a set of premises implies a conclusion (Evans & Grefenstette, 2018a). Using a probability-based estimator, we then empirically extract these rules from the LLM's latent space, instantiating them with features learned by a cross-layer sparse autoencoder (SAE), where each premise is the occurrence of a feature, and a conclusion is the occurrence of another feature. Crucially, after developing a scalable process to categorize these rules by their semantic soundness levels (e.g., strict, plausible, noisy) with a judgment guided by a high-capability LLM, we introduce the *Soundness-Aware Level* (SAL): a quantitative metric, computed via the Jensen-Shannon Divergence, that measures how well a model's internal probability distributions distinguish between sound and unsound rules.

We verify our SAL metric by using it to predict post-RLVR performance across a wide range of pre-trained models from different families (Qwen, Mistral, Llama, DeepSeek) and scales (0.5B to 14B). We observe that a base model with a higher SAL also shows a stronger post-RLVR performance. Quantitatively, we find that a model's post-RLVR error rate ($\epsilon$) can be accurately predicted from its microscopic soundness-aware level ($s$) by an exponential power model: $\epsilon = \exp(-\alpha \cdot s^{\beta})$, which achieves a high fidelity ($R^2 = 0.87$) even for unseen models. This soundness-aware level varies significantly across model families and consistently improves with model scale. These findings provide strong evidence for our central hypothesis: pre-trained models whose internals can already distinguish sound from unsound rules are the most fertile ground for developing strong reasoners via RLVR. This work thus positions SAL as a powerful predictive signature, offering a quantitative, mechanistically-grounded tool for selecting and designing the next generation of reasoning models.

## 2 UNCOVERING THE SIGNATURE OF REASONING

In this section, we develop our framework for discovering a microscopic signature that predicts a pre-trained LLM's reasoning potential. We treat this as a three-step investigation into the model's internal logic. **First**, our approach decodes the raw hidden activations into a set of meaningful features, providing us with the fundamental clues of the model's reasoning. **Second**, we discover the implicit logical rules the model has learned by analyzing the co-occurrence patterns between these features, revealing the connections it has formed. **Finally**, we assess the quality of this learned knowledge by measuring how well the model separates its sound rules from its unsound ones. And we define a single predictive score, the Soundness-Aware Level (SAL), that reveals the mystery of what distinguishes a high-potential reasoner. Figure 1 illustrates this entire process.

## 2.1 Framing Internal Reasoning as Logic Programming

To formally describe the internal reasoning process, we turn to the classic notion from formal logic: that reasoning is the process of repeatedly applying rules (Quine, 1986; Hintikka & Sandu, 2007). Our work is directly inspired by recent research showing that the operations within a transformer can be conceptualized as a system of neural logic. Specifically, Chen (2023) demonstrates that transformer layers can be interpreted as a process of forward-chaining learnable Horn clauses, providing a direct link between deep learning and logical deduction. Separately, another line of interpretability research provides a complementary mechanical intuition for how rule-like behavior can emerge from the feed-forward networks (FFNs) within each block. These studies frame the FFNs as a vast key-value memory system, where the first layer detects feature patterns (keys) and the second layer writes corresponding updates (values) to the residual stream (Geva et al., 2021; Wang et al., 2022). This "if-detect-then-write" operation serves as a powerful mechanical analogue to an "if-premise-then-conclusion" rule. Merging these two perspectives, we adopt the Horn clause as the formal representation for the microscopic reasoning steps we aim to extract and analyze.

To formalize these rules, we adopt the notation of Horn clauses from logic programming (Horn, 1951; Evans & Grefenstette, 2018a). A Horn clause is a specific type of "if-then" statement. In our context, each term in a rule is a feature $c$ discovered by the SAE, and we define an *atom*, $\alpha_c$, to be a boolean variable indicating the activation of that feature (i.e., $\alpha_c = \text{occur}(c)$). A rule with $M$ premises (the body) and one conclusion (the head) is then expressed as:

$$\alpha_{c_1} \wedge \cdots \wedge \alpha_{c_M} \rightarrow \alpha_{c_q}. \tag{1}$$

Here, the conjunction (i.e., $\wedge$) of atoms on the left is the premise of the clause, and the single atom on the right is the conclusion. The rule states that if *all* premise features are active, the conclusion feature should also become active. For example, the model having learned $\sqrt{4} = 2$ could be represented by a rule like: $\text{occur}("\sqrt{}") \wedge \text{occur}("4") \rightarrow \text{occur}("2")$. This formalism allows us to treat the connections between features as a system of logic, which we can then extract and analyze.

## 2.2 Decoding Representations into a Set of Interpretable Features

To analyze a model's internal logic, we must first decode its uninterpretable hidden states into a set of meaningful and interpretable features. To achieve this, we adopt existing works of sparse autoencoders in mechanistic interpretation (Lindsey et al., 2024a), with a focus on the variations for extracting features across different layers. The cross-layer sparse autoencoder is trained to reconstruct each layer's hidden state $\mathbf{x}^l$ using a small number of features activated in that layer or any preceding ones. A sparsity penalty in its loss function (see Appendix B for the full formalism) encourages the model to discover the most efficient and semantically coherent features that explain the LLM's representations. The resulting sparse features are highly interpretable. Following established practice (Cunningham et al., 2023b; Bills et al., 2023), we assign each feature a semantic label ($\mathcal{I}_c$) by prompting a high-capability LLM to summarize the text passages that maximally activate it. To this end, we collect a set of interpretable features (e.g., "the concept of square roots," "coordinate values"), which serve as the atomic units for the rule extraction and soundness analysis that follow.

## 2.3 Discovering Implicit Rules from a Pre-Trained Model

With a set of interpretable features in hand, the next challenge is to discover the rules the model has learned between them. One could attempt this via causal intervention, perturbing features to see their effect on others (Lindsey et al., 2025a). However, this approach is fundamentally ill-suited for discovering logical Horn clauses. For instance, while deactivating the premise features for "$\sqrt{}$" and "4" should stop the conclusion "2" from activating via this specific rule, it should not prevent activating "2" from another valid rule like "$1 + 1$". Perturbation struggles with this "many-to-one" nature of logical entailment and is also computationally prohibitive at scale (Ameisen et al., 2025a).

Therefore, we propose a more robust and scalable probability-based approach that estimates rule strength from feature co-occurrence across a large dataset. The intuition is simple: if a set of premise features $P$ consistently activates in the layers preceding a conclusion feature $Q$ across thousands of varied inputs, this provides strong evidence for a learned rule $P \rightarrow Q$. To formalize this, we define the activation of a feature $c$ for an input $x_n$ at layer $l$ as an atom $\alpha_c^{(n,l)}$, a Bernoulli random variable

that is true (or 1) if the feature's activation $\mathbf{h}_c^l(x_n)$ exceeds a threshold $\tau$. To estimate a rule's probability, we process a dataset of $T$ inputs and compute two co-occurrence statistics:

$$\text{count}(P) = \sum_{n=1}^{T} \left[ \sum_{c_i \in P} \alpha_{c_i}^{(n)} > 0 \right], \qquad \text{count}(P, Q) = \sum_{n=1}^{T} \left[ \sum_{c_i \in P} \alpha_{c_i}^{(n)} > \alpha_{c_q}^{(n)} \right], \qquad (2)$$

where $[\cdot]$ is the Iverson bracket, $c_q$ is the only conclusion feature in $Q$, and $\alpha_c^{(n)} = \sum_{l=1}^{L} \alpha_c^{(n,l)}$ is the total number of times feature $c$ activates for a given input across all layers. From these counts, the conditional probability of the rule is estimated via maximum likelihood with smoothing:

$$\hat{p}(Q \,|\, P) = \frac{\text{count}(P, Q) + \beta}{\text{count}(P) + 2\beta}, \qquad (3)$$

where $\beta$ is a smoothing hyperparameter (e.g., $\beta = 1$ for a uniform prior) that prevents overconfidence when the premise is rarely observed (Murphy, 2012). Intuitively, this equation estimates how often the conclusion is true when the premise is true. Note that the computation uses only feature activations and does not rely on perfect latent-space reconstruction, which is unattainable in practice. This scalable method allows us to extract millions of candidate rules and their corresponding probabilities, forming the raw material for our soundness analysis in the next section.

## 2.4 QUANTIFYING KNOWLEDGE SOUNDNESS-AWARE LEVEL

Having extracted implicit rules from an LLM's internals, our final step is to assess their quality. Our core hypothesis is that a model's reasoning potential is encoded in its ability to distinguish high-quality, logically sound rules from low-quality, spurious ones. To measure this ability, we first categorize the extracted rules into three soundness levels based on their semantics: Strict (representing necessary truths like mathematical theorems), Plausible (representing strong but not universally true heuristics), and No (representing spurious correlations). This judgment is performed solely based on the rules' semantics, where the annotator labels each rule according to the textual explanations of its constituent features. Ideally, these semantic judgments would be provided by human annotators, but the scale of extracted rules makes this infeasible. We therefore use a high-capability LLM as a scalable surrogate for human evaluation (see Appendix D for details).

With rules sorted by soundness, we now formally quantify how well the model separates them. Our goal is to measure the model's aggregate behavior for each category. For instance, a strong model should consistently assign high probabilities to its "Strict" rules and low probabilities to its "Noise" rules. To capture this, we move from analyzing individual rule probabilities to comparing their collective distributions. We effectively create a "confidence histogram" for each soundness category to visualize its overall probability landscape. This is formalized as follows. For each category $y \in \mathcal{Y} = \{\text{Strict, Plausible, Noise}\}$, we gather the set of its transition probabilities, $\mathcal{S}_y = \{\hat{p}(Q \mid P) \,|\, \text{type}(P, Q) = y\}$. To build the histogram, we partition the $[0, 1]$ probability range into $B$ uniform bins. By counting the number of probabilities $n_{y,b}$ from $\mathcal{S}_y$ that fall into each bin $b$, we obtain a normalized probability density function $\boldsymbol{\rho}_y = (\rho_{y,1}, \dots, \rho_{y,B})$, where $\rho_{y,b} = n_{y,b}/|\mathcal{S}_y|$. We then measure the total separation between these distributions using the Jensen-Shannon Divergence (JSD) (Nielsen, 2019a), which we define as our Soundness-Aware Level (SAL):

$$\text{SAL} := \text{JSD}(\{\boldsymbol{\rho}_y\}_{y \in \mathcal{Y}}) = \frac{1}{|\mathcal{Y}|} \sum_{y \in \mathcal{Y}} \text{KL}(\boldsymbol{\rho}_y \,\|\, \boldsymbol{m}), \qquad (4)$$

where $\boldsymbol{m}$ is the mean distribution and $\text{KL}(\cdot\|\cdot)$ is the Kullback-Leibler divergence. A higher SAL score signifies that a model's internal probability assignments for strict, plausible, and noisy rules are markedly different. Essentially, it has learned to separate its high-quality knowledge from its internal noise. This scalar signature is the central predictor we evaluate in the following section.

## 3 EXPERIMENTS: FROM MICROSCOPIC RULES TO REASONING POTENTIAL

This section presents the empirical evidence for our central hypothesis. We begin by visualizing the distributions of internal logic rules, revealing clear structural differences in how strong and weak models treat rules of varying soundness. We then demonstrate that these structural differences

constitute a powerful predictive signature of reasoning potential. This predictive relationship is clear enough to be modeled by an exponential power model that connects our microscopic metric to macroscopic task performance. Finally, we deconstruct this signature by analyzing its relationship with model scale and family, and ground our findings with case studies of the extracted rules.

## 3.1 EXPERIMENT SETTINGS

**Corpus for SAE Training.** To analyze the internal reasoning of each LLM, we first construct a specialized corpus to train our cross-layer Sparse Autoencoders (SAEs). The goal is to create a dataset rich with varied mathematical topics across different difficulty levels. The process begins by curating a foundation of diverse problems from established benchmarks, including the full Math (Hendrycks et al.) dataset, Open Reasoner Zero (Hu et al., 2025), GSM8K (Cobbe et al., 2021), and the AOPS, AMC, and Olympiad subsets from the NuminaMath (LI et al., 2024). After de-duplication, this resulted in a total of 128K unique mathematical questions. Following standard protocols (DeepSeek-AI, 2025; Hu et al., 2025), we then prompt each candidate LLM to generate a "think" style response for every question. This step yields a unique and model-specific corpus for each LLM, consisting of both the questions and the model's own generated reasoning traces, which we then use for SAE training. This corpus will also be used to extract logic rules as detailed in Appendix D. Please note that the **training corpus does *not* include any ground-truth labels**.

**Language Models Under Analysis.** To test our hypothesis across different model scales and families, we select a diverse set of pre-trained LLMs. To analyze the effect of **model scale**, we focused on the high-performing Qwen-2.5 family, including its 0.5B, 1.5B, 7B, and 14B variants (Hui et al., 2024). To examine the impact of **model family**, we then selected three other public models at a comparable $\approx$7B scale: Mistral-7B-v0.1 (Jiang et al., 2023), Llama-3.1-8B (Dubey et al., 2024), and the specialized DeepSeek-Math-7B (Shao et al., 2024).

**SAE Training and Rule Extraction.** To enable a fair microscopic comparison across different LLMs, we apply a consistent protocol to train a dedicated cross-layer SAE for each model on its residual stream. All SAEs share a uniform architecture with $C = 2^{15}$ features and are trained on $L = 8$ layers selected as evenly as possible. For example, in the 28-layer `Qwen-2.5-7B` model, we select every fourth layer for analysis. For the training process, we follow established best practices (Lindsey et al., 2024b; Gao et al., 2024), using the AdamW optimizer (Loshchilov & Hutter, 2017) with standard parameters ($\beta_1 = 0.9, \beta_2 = 0.999, \epsilon = 6.25 \times 10^{-10}$). We use a learning rate of $2 \times 10^{-4}$ with a cool-down in the final 20% of steps, and a sparsity penalty $\alpha$ of $5 \times 10^{-3}$ with a linear warm-up over the first 20% of steps. Our trained SAEs balance reconstruction fidelity with feature sparsity, yielding a relatively low normalized MSE of 0.65-0.80 while using an average of only 20-30 active features to reconstruct each token's representation. Full details are in Appendix C. To ensure meaningful analysis, we further focus on the SAE features that demonstrate reliable monosemantic explanations, as detailed in Appendix C.2. Once we obtain trained cross-layer SAEs, we count their co-occurrence probability over a subset of the full training data. More details and engineering efforts for speeding up this process are described in Appendix D.

**Scalable Annotation with LLMs.** To scalably analyze our microscopic findings, we extend a methodology that is now standard practice in LLM interpretability: using high-capability LLMs to generate semantic explanations for internal model features (Bills et al., 2023; Gao et al., 2024). Our process consists of two stages. First, an LLM generates a textual explanation for each individual SAE feature. Second, in a small extension, we use the same LLM judge, DeepSeek-R1 (Guo et al., 2025), to classify the logic rules formed by these features as Strict, Plausible, or Noise. The task of judging rule soundness is inherently challenging. We assess the reliability of this automated labeling process against human judgments in Appendix C.2. However, the ultimate validation of our method is not the label agreement score, but the predictive power of the final SAL metric against macroscopic task performance. Notably, while the labels are necessarily noisy, we find that the SAL metric derived from them is a robust predictor of actual reasoning accuracy, as demonstrated in our main results in the following subsections. This suggests the process captures a strong underlying signal, proving robust to the inherent noise of the intermediate soundness labels.

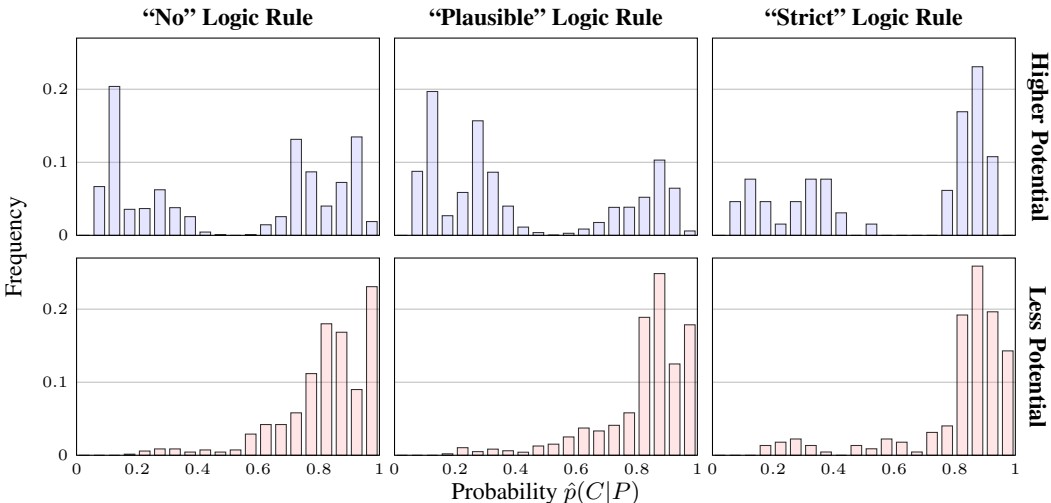

Figure 2: Probability distribution of extracted logic rules from a higher-potential (`Qwen-2.5-7B`) model and a lower-potential (`Llama-3.1-8B`) model. The higher-potential model shows significantly different distributions for No, Plausible, and Strict logic rules, whereas the lower-potential model collapses toward similar shapes, indicating a failure to recognize different soundness levels.

## 3.2 Microscopic Differences Between Models with High and Low Potential

**Key Finding: Stronger models visually and quantitatively separate their internal rules by soundness, while less-potential models do not.** We begin by visualizing the internal rule distributions, which reveal a stark difference between models with high and low reasoning potential. Figure 2 provides the direct visual evidence. The stronger model, `Qwen-2.5-7B` (top row), is clearly soundness-aware: it exhibits three qualitatively different confidence histograms.

Its "Strict" rules cluster tightly at high probabilities ($> 0.8$), its "Plausible" rules form a broad mid-range distribution, and its "Noise" rules are correctly concentrated at low probabilities. In contrast, the less-potential model, `Llama-3.1-8B` (bottom row), is soundness-agnostic. It shows nearly identical, right-skewed distributions for all three soundness levels. Most of its rules, regardless of their actual soundness, are assigned a high probability. This suggests that the less-potential model treats most feature co-occurrences as equally reliable, blurring the logical boundaries that a more capable reasoner maintains.

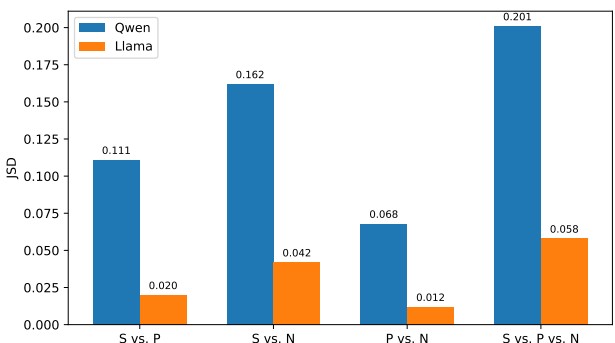

Figure 3: Jensen-Shannon Divergence (JSD) quantifies distribution shifts between probabilities of soundness levels (S: "Strict", P: "Plausible", & N: "No").

To formalize this visual gap, we quantify the separation between these distributions using Jensen-Shannon Divergence (JSD). As shown in Figure 3, the stronger model achieves a high overall SAL score of 0.201, while the less-potential model's score is a much lower 0.058. This confirms that SAL effectively captures the qualitative difference in the models' internal knowledge structure, serving as a reliable indicator of soundness-awareness.

## 3.3 The Predictive Power and Generality of the SAL Metric

**Key Finding: SAL is a robust predictor of post-RLVR performance, characterized by an empirical correlation.** Figure 4 (left) plots each model's SAL score against its average post-RLVR accuracy. The points indicate a strong monotonic relationship between our microscopic signature

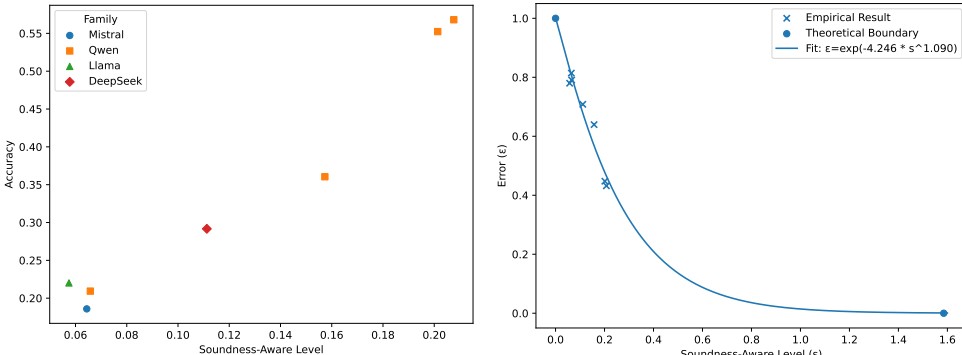

Figure 4: Left: Correlation between the SAL over extracted rules and models' post-RLVR performance. Right: The exponential power correlation $\epsilon = \exp(-\alpha \cdot s^\beta)$ describes the correlation between SAL $s$ and the error rate $\epsilon$ of solving mathematical problems. The best fitted model is $\alpha = 4.246$ and $\beta = 1.090$ with $R^2 = 0.985$ for interpolation fitting observed models.

Table 1: Spearman correlation (%) between SAL and model performance after RLVR training. The post RL (GRPO) performances of different base models are referenced from Zeng et al. (2025a). Statistical significance is denoted by ** and * for p-values < 0.01 and < 0.05, respectively.

| Metric | AMC23 | MATH500 | Minerva | Olympiad | AIME24 | Avg. Acc. |
|---|---|---|---|---|---|---|
| *#Behavior* | | | | | | |
| Verification | 96.43** | 85.71* | 75.00 | 85.71* | 77.83* | 85.71* |
| Backtracking | 85.71* | 67.86 | 50.00 | 67.86 | 63.01 | 67.86 |
| Subgoal | 89.29** | 78.57* | 67.86 | 78.57* | 77.83* | 78.57* |
| Backward | 3.60 | 5.41 | 9.01 | 5.41 | 26.18 | 5.41 |
| *Pre-RL Perf.* | | | | | | |
| GSM8K | 85.71* | 85.71* | 75.00 | 85.71* | 81.54* | 85.71* |
| MATH500 | 92.86** | 100.0** | 96.43** | 100.0** | 96.36** | 100.0** |
| *Ours* | | | | | | |
| SAL | 89.29** | 96.43** | 92.86** | 96.43** | 96.36** | 96.43** |

(SAL) and macroscopic performance. Models with small SAL scores ($< 0.08$) achieve only 20% accuracy, while models with the highest SAL scores ($> 0.20$) see their performance more than doubled. This pattern provides compelling evidence that larger separations in a model's internal rule distributions coincide with better reasoning potential.

We empirically model this correlation to confirm this relationship and test its forecasting ability. We anchor this correlation with two *hypothesized* theoretical boundaries: a model with zero ability to distinguish rules (SAL = 0) should have a 100% error rate ($\epsilon = 1$), and a hypothetical perfect model (SAL = $\log_2(3)$) should achieve a 0% error rate ($\epsilon = 0$). Using these anchors, we fit an exponential power model for the data points, a functional form inspired by large deviation theory, which connects the probability of rare events to the divergence between distributions (Cover & Thomas, 2006):

$$\epsilon = \exp(-\alpha \cdot \text{SAL}^\beta). \tag{5}$$

This correlation, with fitted parameters $\alpha = 4.25$ and $\beta = 1.09$, captures 98.5% of the variance ($R^2 = 0.985$) in the observed data. To confirm its generalization, we conducted a leave-one-out validation, which successfully forecasted the performance of held-out models with $R^2 = 0.872$.

Finally, to situate SAL's performance, we compare it against other predictive metrics using Spearman correlation across multiple benchmarks (Table 1). SAL achieves a high average correlation (96.4%) and consistently outperforms behavioral metrics. While pre-RL accuracies on benchmarks like MATH500 and GSM8K are also strong predictors, they share a critical limitation: they require a large, labeled, in-domain dataset to compute. In contrast, SAL is a **zero-label metric** with respect to the downstream task. It is derived solely from the internal statistics of a pre-trained model on an unlabeled corpus, using only intermediate semantic labels from an LLM judge, not ground-truth problem solutions. This makes SAL a more fundamental intrinsic signature of reasoning potential.

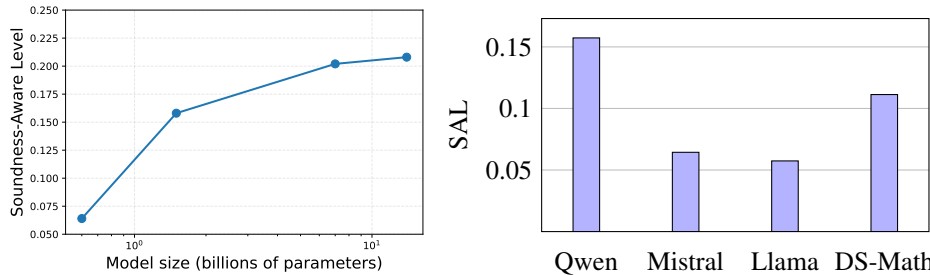

Figure 5: Deconstructing the Soundness-Awareness Level (SAL). **(Left)** SAL increases with model scale within the Qwen-2.5 family. **(Right)** At a comparable 7B scale, SAL varies significantly across different model families, indicating that architecture and pre-training data are also key factors.

### 3.4 THE IMPACT OF MODEL SCALE AND FAMILY ON SOUNDNESS-AWARE LEVEL

**Key Finding: Soundness-aware level increases with model scale and varies significantly across model families.** Our analysis reveals that SAL is strongly influenced by both model scale and family. First, we find that micro-level differentiation grows monotonically with model scale. Figure 5 (Left) shows that within the Qwen family, SAL climbs from around 0.06 in the 0.5B model to around 0.22 in the 14B model, with a clear upward trajectory. The increase is sharp at smaller scales but becomes more moderate beyond 1.5B, suggesting diminishing returns. Beyond 14B, the curve appears to approach saturation, indicating that additional parameters may mostly serve to refine existing rule clusters rather than create new separations. In other words, while capacity helps the model sort rules more cleanly up to a threshold, the marginal gains of simply scaling further begin to taper. This pattern suggests that future work on architectural or data interventions may be more effective than scaling alone for improving this core reasoning potential. While scale is a clear factor, model family, which encapsulates differences in architecture and pre-training data, plays an equally critical role. Figure 5 (Right) compares four models at the 7B scale, revealing high variations in SAL. Qwen scores highest at approximately 0.16 and the specialized DS-Math reaches roughly 0.11, whereas the more generalist Mistral and Llama models stay near 0.06. Since the parameter count is fixed, these gaps demonstrate that a model's family leaves a recognizable microscopic signature that promotes or limits the separation of its learned rule distributions, and by extension, its reasoning potential.

### 3.5 CASE STUDIES

Table 2 provides a direct look at the kinds of rules `Qwen-2.5-7B` has learned. These case studies reveal a clear hierarchy in the model's internal logic, where the semantic quality of a rule corresponds directly to the confidence the model assigns it. For what the model treats as a "Strict" rule ($p \approx 0.98$), we find a near-deterministic pattern: the presence of an equivalence symbol (`\equiv`) with a variable (`$a`) reliably signals an algebraic equation. For a "Plausible" rule ($p \approx 0.90$), we see a strong procedural heuristic: phrases for isolating a variable (e.g., "solve for x") consistently precede the operation "divide both sides." Finally, for a "Noise" rule ($p \approx 0.29$), the model correctly assigns a very low probability to a spurious link between LaTeX delimiters and a generic phrase. Notably, these examples highlight that even the model's "strictest" rules are not formal logical theorems but are instead reliable contextual deductions learned from the data. This underscores the inherently probabilistic nature of knowledge in LLMs. The key finding is not that the model has learned perfect logic, but that it has successfully learned to organize its knowledge into a hierarchy of reliability. It internally separates its near-deterministic deductions from its useful heuristics and its spurious patterns by assigning them markedly different probabilities. This is the very phenomenon of soundness-awareness that our SAL metric is designed to capture.

## 4 RELATED WORKS

Recent efforts to understand reasoning in LLMs span multiple perspectives. From a behavioral view, studies analyze cognitive habits that correlate with self-improving reasoning. These include explicit phrases related to cognitive behaviors like verification, backtracking, and subgoal decomposition, with findings that stronger models tend to generate a more diverse set of such behaviors (Gandhi et al., 2025; Yue et al., 2025a; Cai et al., 2025; Li et al., 2025). Structural analyses instead model the

Table 2: Examples of extracted logic rules. For each case, we report its $\hat{p}(C|P)$ along with the feature explanations ($P_1$, $P_2$, and $C$), the soundness level, and the rationale (all by DeepSeek-R1).

| | |
|---|---|
| ***Example 1:* "Strict",** $\hat{p}(C|P_1 \wedge P_2) = 0.9766$ | |
| $P_1$ | Pattern "\equiv". |
| $P_2$ | Pattern "\$a" as a variable (e.g., "length \$a", "integer \$a", "coordinates ... (-a", "2a"). |
| $C$ | Algebraic equations and expressions ending with numerical results (e.g., "= 0", "= 16", "$\hat{2}$"). |
| Justify | Equivalence relations with variable "a" imply algebraic equations. |
| ***Example 2:* "Plausible",** $\hat{p}(C|P_1 \wedge P_2) = 0.8960$ | |
| $P_1$ | Steps in mathematical problem-solving involving solving/calculating variables or terms, e.g., "solve for (x)", "calculate (c)", "find tan(B)". |
| $P_2$ | Pattern "which" in math explanations (e.g., "values of (b) for which", "integer $x$ for which"). |
| $C$ | Pattern "divide both sides". |
| Justify | Using division to isolate variables under conditions. |
| ***Example 3:* "Plausible",** $\hat{p}(C|P_1 \wedge P_2) = 0.8461$ | |
| $P_1$ | Pattern "formula". |
| $P_2$ | Mathematical expressions with addition in algebraic equations, e.g., "$x^2$ +", "x +", "$ax^3$ +". |
| $C$ | Pattern "Numerical value" or equation followed by a period (e.g., "X.", "Y/Z.", "= Z\$."). |
| Justify | Algebraic steps with addition may lead to numerical solutions. |
| ***Example 4:* "No",** $\hat{p}(C|P_1 \wedge P_2) = 0.2854$ | |
| $P_1$ | Start of mathematical expressions/equations in LaTeX, e.g., "[...", "\$...", "sum", "sqrt", "frac". |
| $P_2$ | Pattern "\$" indicating LaTeX math mode initiation/termination. |
| $C$ | Pattern "According to the problem". |
| Justify | No logical or heuristic link between LaTeX math and problem reference. |

"thinking process" as a graph, showing that stronger models produce reasoning graphs with richer cyclic structure (Minegishi et al., 2025). Other work investigates the role of uncertainty and model confidence, using metrics like Pass@K and Entropy to emphasize how RLVR continually increases the model's confidence in the correct answer (Wen et al., 2025; Cui et al., 2025; Zeng et al., 2025b; DeepSeek-AI, 2025; Yue et al., 2025b). Mechanistic interpretation approaches move beyond outputs to understand model internals. Sparse autoencoders and cross-layer transcoders recover semantically meaningful features and the circuits they form, revealing how multi-step deductions emerge inside transformers (Cunningham et al., 2023a; Bricken et al., 2023b; Ameisen et al., 2025a). Early explorations in this direction have built causal graphs of these features to perform case studies on specific behaviors like multi-hop reasoning, though these have remained largely qualitative (Lindsey et al., 2025b; Ameisen et al., 2025b). Inspired by recent progress in logic programming, where they frame reasoning as rule application and linking neural features to formal inference systems (Evans & Grefenstette, 2018b; Csiszár, 1975; Nielsen, 2019b; Chen, 2023), we consider our microscopic view of reasoning as distributions of logic rules extracted from internal representations.

## 5 CONCLUSION

We introduced the Soundness-Awareness Level (SAL), a novel microscopic signature that provides a reliable predictive signal for the downstream reasoning potential of pre-trained language models after RL training. Our framework moves beyond analyzing macroscopic behaviors, instead extracting the implicit logical rules a model has learned and quantifying its intrinsic ability to distinguish sound knowledge from less sound ones. Our experiments demonstrate that SAL is a powerful predictor, whose relationship with macroscopic error rates can be characterized by an exponential power model ($R^2 = 0.87$). Furthermore, as a zero-label metric, SAL offers a more fundamental and intrinsic signature of a model's reasoning potential. This work represents a first step toward a more mechanistic approach to understanding reasoning. By providing a quantitative link between a model's internal knowledge structure and its emergent capabilities, SAL not only offers a practical tool for model selection but also opens new avenues for designing pre-training objectives, architectures, and constructing pre-training datasets that explicitly cultivate soundness-aware abilities from the start.

**Limitations and Future Work.** While our work establishes a strong predictive correlation, proving a direct causal link between SAL and reasoning potential is a crucial direction for future research. We do not perform the interventional experiments necessary to demonstrate causality. However, our framework provides the foundational metric and strong evidence to motivate such studies. In this work, we consider reasoning primarily through mathematical tasks, and extending SAL to broader forms of reasoning such as coding is an important direction for future exploration.

## 6 ETHICAL STATEMENT

This work analyzes publicly available base models and checkpoints under their respective licenses, and we used them strictly for research. Particularly, our study evaluates multiple families and scales, including Qwen (Hui et al., 2024), Mistral (Jiang et al., 2023), Llama (Dubey et al., 2024), and DeepSeek (Shao et al., 2024), as described in the main text, and it relies on math-focused benchmarks such as MATH (Hendrycks et al.), GSM8K (Cobbe et al., 2021), and NuminaMath subsets (LI et al., 2024) that are broadly used by the research community. We complied with all dataset and model usage terms and did not collect or process any personal data. No human subjects research was conducted, and no personally identifiable information appears in the paper.

## 7 REPRODUCIBILITY STATEMENT

We structure the details of our implementation here to reproduce our results. Sections 2.2 to 2.3 describe our proposed full pipeline. Appendix B and Appendix C provide implementation details for the cross-layer SAEs, including architecture, sparsity objective, and key hyperparameters that we used to train SAEs across model families. Section 3.1 documents datasets, preprocessing, model families and scales, and evaluation protocols, and the machine annotation procedure with prompts and the label taxonomy appears in Appendix C and Appendix D. Algorithmic details for efficient rule extraction and counting are in Appendix D. The computing resources we required to conduct our experiments are described in Appendix E. We will release our code and data to reproduce all results reported in the paper once accepted.

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

## A    LLM USAGE STATEMENT

We leverage LLMs for three distinct purposes, and the terms we applied are as follows:

**LLM as Research Subjects.** This research focus of this paper is on understanding the internal differences between pre-trained models that can be trained to be powerful reasoning models and those that cannot. Therefore, our research considering publicly available LLMs (i.e., Qwen-2.5 (Hui et al., 2024), Llama-3.1 (Dubey et al., 2024), Mistral-v0.1 (Jiang et al., 2023), and DeepSeek-Math (Shao et al., 2024)) following their academic usage policy.

**LLM as Human Annotator.** In some of our experiments, we require scaling up our experimental results by using LLMs to simulate human annotators. In particular, the automatic annotation process is empowered by DeepSeek-R1 (DeepSeek-AI, 2025), and we follow their general user policy.

**LLM for Writing Assistance.** During the writing of this manuscript, we leverage ChatGPT [1] to improve the writing quality by correcting grammar/typo issues, rephrasing the terms for clarity, and providing visualization suggestions for tables and figures. We confirm that all the contents from the manuscript have been manually checked by us, and they represent our original thoughts.

## B    CROSSCODER FORMALISM AND TRAINING DETAILS

We provide the implementation details for the cross-layer sparse autoencoder (SAE) that will be used to extract features from the hidden states of a pre-trained LLM, as mentioned in Section 2.2.

Given an $L$-layer pre-trained LLM that produces residual-stream hidden states $\{\mathbf{x}^l\}_{l=1}^L$ where $\mathbf{x}^l \in \mathbb{R}^D$, we train a Crosscoder $f_{\text{SAE}}$ to recover these states from a sparse feature representation. The Crosscoder consists of $L$ pairs of trainable encoder-decoder weights $\{(\mathbf{E}^l, \mathbf{D}^l)\}_{l=1}^L$, where the encoder and decoder matrices $\mathbf{E}^l, \mathbf{D}^l \in \mathbb{R}^{D \times C}$ and the feature dimension $C$ is much larger than the hidden state dimension $D$ ($C \gg D$).

For each layer $l$, the Crosscoder first encodes the hidden state $\mathbf{x}^l$ into its sparse, non-negative feature space $\mathbf{h}^l = \text{ReLU}(\mathbf{x}^l \mathbf{E}^l) \in \mathbb{R}_+^C$. The decoder then reconstructs the $l$-th layer hidden state, $\hat{\mathbf{x}}^l$, using the feature activations from the current layer and all preceding layers:

$$\hat{\mathbf{x}}^l = \sum_{l'=1}^l \mathbf{h}^{l'} \mathbf{D}^{l\top}.$$

This cross-layer reconstruction allows the model to capture features at their layer of emergence and reuse them for reconstruction in subsequent layers. The Crosscoder is trained by minimizing the following loss function:

$$\mathcal{L} = \sum_{l=1}^L \|\mathbf{x}^l - \hat{\mathbf{x}}^l\|^2 + \alpha \cdot \sum_{l'=1}^L \sum_{c=1}^C \|\mathbf{h}_c^l \cdot \mathbf{D}_{:,c}^{l'\top}\|_1, \tag{6}$$

where $\alpha$ is a hyperparameter controlling the sparsity level. The first term is the mean squared reconstruction error. The second term is an $L_1$ penalty that encourages sparsity. Notably, this sparsity penalty is applied to the feature activation $\mathbf{h}_c^l$ multiplied by its corresponding decoder weights $\mathbf{D}_{:,c}^{l'}$, ensuring that a feature is only penalized when it is actively used for reconstruction. Overall, this objective encourages the model to explain the LLM's hidden states using the fewest possible features.

## C    TRAINING AND INTERPRETING CROSS-LAYER SAEs

### C.1    TRAINING CROSS-LAYER SPARSE AUTOENCODER

Following previous research (Lindsey et al., 2024b), we train our cross-layer SAE on the residual stream of the subject LLMs. For a comparable analysis on extracted features across different models, we design our cross-layer SAE to share the same configuration, with the number of features $C = 2^{15}$ and the total number of layers $L = 8$. We evenly choose the layer to monitor across each candidate

---

[1]ChatGPT is available at: https://chatgpt.com/

model. For example, for `Qwen-2.5-1.5B` and `Qwen-2.5-7B`, having exactly 28 layers in total, we monitor the residual stream of them every four layers, where the first monitor layer is the input of the first layer. For other models whose total number of layers cannot be evenly divided by 8, we manually choose certain layers that are almost evenly divided by 8. Following recommendations by Gao et al. (2024), we apply AdamW (Loshchilov & Hutter, 2017) optimizer with $\beta_1 = 0.9$, $\beta_2 = 0.999$, and $\epsilon = 6.25 \times 10^{-10}$ to train crosscoders using an initialized learning rate $2 \times 10^{-4}$ with a cool down strategy in the last 20% steps as suggested by Lindsey et al. (2024b). The default sparse penalty $\alpha$ is $5e^{-3}$ with a linear warm-up strategy over the first 20% steps. To ensure the training is stable, we apply a relatively large batch size, setting the batch size to 128 for all models. Please note that we count the batch size at the instance level, rather than the token level, resulting in approximately 60,000 tokens per batch. We find that enlarging the training batch size is the most important trick to prevent the failure of training cross-layer SAEs. The training will continue for a total of 5K steps, referring to around 5 epochs on our dataset. The above hyperparameters are applicable to most of the base models, except for the largest candidate `Qwen-2.5-14B`, which requires a relatively larger sparse penalty at $3 \times 10^{-3}$. To this end, the trained crosscoders reach around 0.65-0.80 normalized MSE with an average of $\approx 20$ activated features to reconstruct each token. By feeding data, in Table 3, we observe that only 3.43% of features have not been activated for any input text across all model families, indicating they are dead during the SAE training.

### C.2 Interpreting Cross-layer Sparse Autoencoder

Following previous research (Bricken et al., 2023a), we interpret the semantics of each learned feature vector $c$ from our cross-layer SAEs by collecting the text spans that could maximally activate each instance. In particular, once the SAEs are trained, we feed them with all $128K$ training corpus and then collect the text spans that could maximally activate for each learned feature, regardless of which layer they activate in. We then interpret the semantics of each feature by summarizing the top 15 most activated text spans for each feature. As suggested by previous work Lieberum et al. (2024), we further check the confidence of the summary by assessing whether the same pattern can be observed from the top 30 most activated text spans for each learned feature. Following previous

Table 3: Dead and explainable rates of features discovered by our cross-layer SAE across candidate LLMs. Explainable is the fraction of *sufficiently* activated features labeled as Yes, Probably, or Maybe by the LLM judge, following the protocol in Section C.1.

| Model | Dead Rate | Explainable Rate |
|---|---|---|
| `Qwen-0.5B` | 0.00% | 82.59% |
| `Qwen-1.5B` | 3.17% | 92.61% |
| `Qwen-7B` | 17.98% | 96.27% |
| `Qwen-14B` | 0.34% | 86.48% |
| `Llama-8B` | 2.20% | 95.45% |
| `Mistra-7B` | 0.09% | 76.02% |
| `Deepseek-7B` | 0.21% | 89.14% |
| Avg. | 3.43% | 88.37% |

works (Bills et al., 2023; Lieberum et al., 2024), this annotation process can be *reliably* scaled up with modern LLMs, where we choose DeepSeek-R1 Guo et al. (2025) as our LLM judge. The prompting templates for generating an initial summary and checking the confidence are listed in Figure 7 and Figure 8, respectively. Empirically, we focus on the features that have been activated over 30 times over the entire corpus, resulting in an occurrence probability of 0.02%. We observe that the overall interpretability of these sufficiently activated features matches previous research (Lieberum et al., 2024). For example, 68.34%, 19.11%, 8.81%, and 3.73% of sufficiently activated features from `Qwen-2.5-7B` are labeled with confidence level "Yes", "Probably", "Maybe", and "No", respectively, resulting in a 96.27% overall explainable rate if we consider confidence with "Maybe" or above as effectively explained. These results confirm that our trained cross-layer SAEs can provide interpretable features for our further analysis.

## D Extracting Logic Rules from LLMs

We extract the logic rules with our proposed probability-based estimator in Section 2.3.

**Dataset.** Since collecting the co-occurrence probabilities for all activated features is time-consuming, we can only select a subset of our entire dataset. To ensure that our selected data

can cover all kinds of mathematical concepts and reasoning skills, we construct our dataset from the MATH dataset Hendrycks et al. as it systematically covers 7 categories of mathematical concepts, and provides questions with different difficulties for each concept. Specifically, we randomly sample 100 samples for each difficulty level within each category, resulting in a total of 3267 samples. Following our standard protocol of preparing data as described in Section 3.1, we also consider the generated responses from each candidate model.

**Extracting Details.** Our proposed method can theoretically extract rules with an arbitrary length of the premises. However, it becomes infeasible in our computing budget. Therefore, we focus on capturing rules with 1 more 2 premise features only, which already results in $\binom{32768}{3} \approx 5.8 \times 10^{12}$ possible combinations. In addition, we make three engineering efforts to accelerate the counting process. Firstly, for each input text with $N$ words, instead of counting over token by token, we monitor their feature activations by aggregating them at the last token. It is reasonable for modern LLMs because the model can read feature activations for any preceding layers and preceding tokens, while it cannot read them from upper layers, even from the preceding tokens. On this path, we further simplify the counting process by aggregating the times that have been activated over all layers at the last token. To this end, for the activations of $C$ features over $N$ tokens from an $L$-layers LLMs, we reduce it from a tensor of shape $\mathbb{R}^{N \times L \times C}$ to a vector of $\mathbf{R}^C$, where each dimension counts how many layers the corresponding feature is activated. Secondly, we implement two specific operators to count the co-occurrence frequency with 1 or 2 features as premises, respectively. In particular, we can vectorize the conditional counting by comparing the number of activated layers. That is, if a feature is activated over $l_1$ layers in total, and another feature activated over $l_2$ layers, then we have $\text{count}(c_1, c_2) + = 1$ if $l_1 > l_2$. Algorithm 1 and 2 demonstrate these two engineering details. Thirdly, we parallel this counting process over multiple computing nodes and aggregate their final counting results once they all finish. To further speed up the counting process, we focus on those features whose text explanations are verified at least "Maybe" level according to the protocol described in Appendix C.2. Empirically, counting the co-occurrence of features over those 3267 samples requires around 30 hours for each model on one single node, and this process can be reduced linearly over the number of available nodes.

**Annotation Details.** To scale up the annotation of soundness levels for our extracted rules, we use one of the most capable LLM, i.e., DeepSeek-R1 (DeepSeek-AI, 2025), with its thinking mode enabled. For reproducibility, we choose a relatively low temperature for generation, where we set $temperature = 0.1$ and $top_p = 0.9$. In addition, we do not restrict the generation length to ensure that we do not limit its strong reasoning capability for annotating the soundness levels of the rules. We present the prompting templates that we used for this automatic annotation process in Figure 9 and Figure 10 for extracted rules with single or multiple horn clauses, respectively. The outputs of the annotation process will be in a valid JSON format. We will then parse the JSON and identify the soundness level identified by DeepSeek-R1, along with the rationale provided.

**Annotation Quality.** To confirm whether DeepSeek-R1 (DeepSeek-AI, 2025) can perform reliable annotations on the soundness levels of the extracted logic rules, we perform a small human study. In particular, we randomly select 30 extracted logic rules from `Qwen-2.5-1.5B` and another 30 from `Qwen-2.5-7B`, and one of the authors will be responsible for assigning a human label to their soundness level. We observe that the overall agreement between DeepSeek-R1 and human annotators is 0.566 for this task. By extending our qualitative analysis, we find that DeepSeek-R1 occasionally confuses "Strict" rules with "Plausible" ones or "Plausible" rules with "No" ones, but it rarely mistakes "Strict" rules for "No" rules. It suggests that most disagreements arise along the softer boundaries between adjacent categories, rather than the harder ones. Although higher-quality or larger-scale human annotations may further improve the precision of these labels, we find the current annotation quality sufficient for the aggregate-level analysis performed in this work.

## E  COMPUTING RESOURCES

We implement our proposed methods with Pytorch, and we run all experiments on no more than three computing nodes with the following configuration. Each node is equipped with 96 virtual CPU cores, 1 TB of main memory, 8 TB of cloud-connected disk space, and 8 A100 Nvidia GPUs with 80GB of GPU memory each. To train the cross-layer SAEs for our largest candidate LLM (i.e., `Qwen-2.5-14B`), this single node requires training for around 60 hours. To extract the logic

rules with the trained SAE for it, it takes about 50 hours and requires a maximum of 500GB of main memory, while it does not require too many GPU resources. We observe that the time of extracting logic rules can be linearly reduced by increasing the available computing nodes.

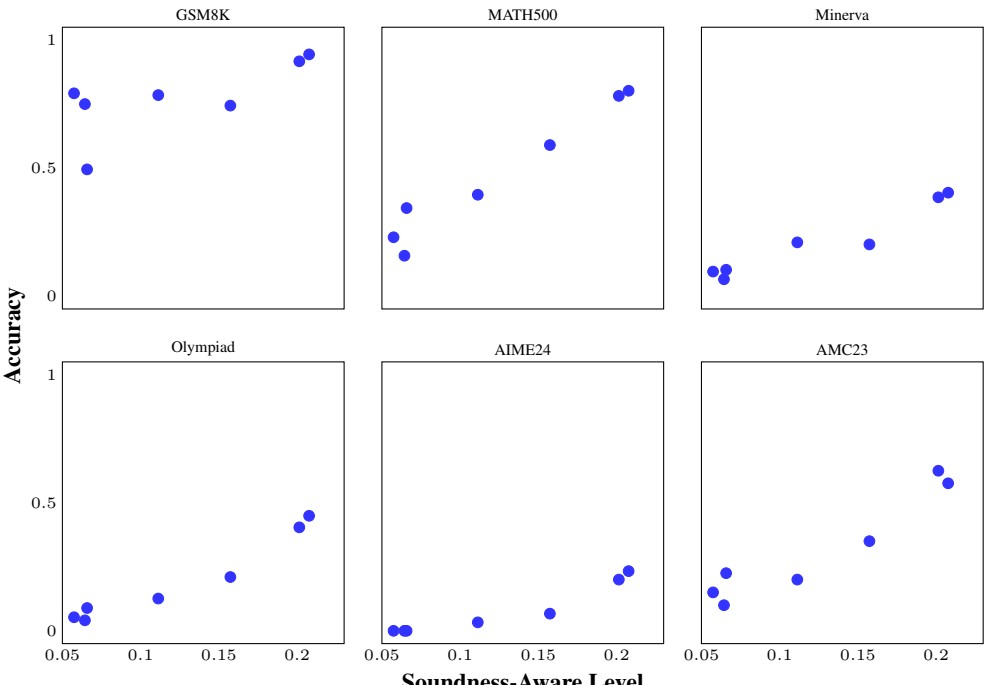

Figure 6: Correlation between soundness-aware level and post-RLVR accuracy for each dataset.

---

**Algorithm 1** Feature Activation Aggregation:

---

**Require:** $x$ with `len(x.shape) == 3`      $\triangleright x \in \mathbb{R}^{L \times N \times C}$
**Require:** `feat_idx, threshold` $= T$
1: $x \leftarrow \text{cumsum}(x, \text{axis} = 0)[-1]$      $\triangleright x \in \mathbb{R}^{N \times C}$
2: $x \leftarrow \big(x[:, \text{feat\_idx}] > T\big).\text{bfloat16}()$    $\triangleright x \in \{0, 1\}^{N \times C'}$
3: $x \leftarrow \text{cumsum}(x, \text{axis} = 0)[-1]$      $\triangleright x \in \mathbb{N}^{C'}$
4: **return** $x$     $\triangleright$ length-$C'$ vector of per-feature counts

---

---

**Algorithm 2** Vectorized Update for $P = 1$ and $P = 2$ Premises

---

**Require:** $x$: vector of layer counts per feature (length $C$).
1: **// Initialize**
2: `Counts = {}`
3: $A \leftarrow \{\, c : x[c] > 0 \,\}$

4: **// Record for feature occurring probability.**
5: **for all** $p \in A$ **do**
6:     `Counts[(p,)][-1] += 1`
7: **end for**

8: **// 1-Premise Count** ($p \Rightarrow q$)**, vectorized masks from** $x$
9: `pair` $\leftarrow$ `(x>0)[:,None]` $\wedge$ `(x>0)[None,:]`    $\triangleright$ $C \times C$
10: `smaller` $\leftarrow \big(x[\text{None},:] < x[:,\text{None}]\big) \wedge$ `pair`
11: $(\text{prem}, \text{concl}) \leftarrow$ `NonZero(smaller)`
12: **for** $i \leftarrow 1$ **to** `len(prem)` **do**
13:     $p \leftarrow \text{prem}[i], q \leftarrow \text{concl}[i]$
14:     `Counts[(p,)][q] += 1`
15: **end for**

16: **// 2-Premises Counts** ($p_1 \wedge p_2 \Rightarrow q$)
17: `prod` $\leftarrow$ `einsum("ac,bc->abc", smaller, smaller)`    $\triangleright$ $C \times C \times C$
18: $(r, c) \leftarrow$ `LowerTriangularIndices(C)`
19: `prod[r, c, :]` $\leftarrow 0$    $\triangleright$ enforce $p_1 < p_2$, drop diagonals
20: $(p_1, p_2, q) \leftarrow$ `NonZero(prod)`
21: **for** $i \leftarrow 1$ **to** `len(`$p_1$`)` **do**
22:     `Counts[(p_1[i], p_2[i])][q[i]] += 1`
23: **end for**

---

> We are studying the behaviors of neurons from a language model. Looking at some text spans activated by the neuron and summarize what feature the neuron is looking for. Please pay most attention to __the ending of each span__. Your summary should be in one (short) sentence, and only describe the most significant feature.
>
> Organize your final summary within the special tag: <summary> summary here </summary>. If there is one short lexical pattern duplicated across all text spans, you extract it out: <summary> Exact pattern: "Key Pattern" with context info here </summary>. An extracted pattern typically is a single word/phrase ("xxx", where "xxx" can be a specific word or pattern), an n-gram (e.g., "xxx yyy" or "xxx yyy zzz"), or a skip-gram ("xxx ... yyy"). If there is no lexical patterns shown off, try to summarize the semantic of the text spans: <summary> Semantic: semantic behind the spans, with a few "Examplar Patterns" here </summary> The semantic usually is a specific concept, topic, or theme, expressing by multiple semantic similar phrases (e.g., "... xxx/yyy/zzz ...", where xxx/yyy/zzz semantically share the same concept). If you cannot summarize the text spans with a clear pattern or concise semantics, you should say: <summary> Cannot Tell </summary>.
>
> Keep your <think> as short as possible, don't repeat your think again and again.
>
> The following are text spans that can maximally activate a certain neuron:
> Span 1: [[ Insert Span 1 Here ]]
> Span 2: [[ Insert Span 2 Here ]]
> ...

Figure 7: Prompting template for summarizing the semantics of learned feature vectors. Note that, since we use DeepSeek-R1 as our judge model, we put this instruction directly in the User role instead of the System part as suggested by the official document.

You are a linguistic expert.

Determine whether the given feature is fuzzy matched by the txt spans. *Fuzzy Matched* means the *semantic/concept* of the given feature is explicitly or *implicitly* shown.

Organize your final decision in the format of "Final Decision: [[ Yes/Probably/Maybe/No ]]". "Yes/Probably/Maybe" indicates at least 85%/65%/40% text spans include the given feature.

Keep your <think> as short as possible, don't repeat your thought again and again.

Feature: [[ Insert the Feature Summary Here]]
Span 1: [[ Insert Span 1 Here ]]
Span 2: [[ Insert Span 2 Here ]]
...

Figure 8: Prompting template for checking the confidence of generated summary. Note that, since we use DeepSeek-R1 as our judge model, we put this instruction directly in the User role instead of the System part as suggested by the official document.

**Task**
For the given premise $P$ and conclusion $C$, judge whether the implication
$$
P \to C
$$
is a **Strict or Plausible Horn Clause**, i.e., whether the occurrence of premise $P$ in the front can point toward the occurrence of conclusion $C$ later in solving mathematical problems. You should classify the given horn clause candidate into one of "Strict/Plausible/No". Here, "Strict Horn Clauses" capture causally and logically relations (e.g., mathematical theorems), while "Plausible Horn Clauses" capture helpful intuitions or heuristic strategies to solve math problems (e.g., planning and checking) *without* strict logical soundness required. If the horn clause candidate does *not* reflect any strict relations or plausible intuitions, classify it as "No", indicating not a horn clause.

**Premise ($P$)**
[[ Insert Premise Here ]]

**Conclusion ($C$)**
[[ Insert Conclusion Here ]]

**Output**
Organize your final judgement as a **JSON** object with the following keywords:
"Category": select from "Strict/Plausible/No", "Relation/Intuition": a string less then 25 words
The JSON object should be wrapped by a special tag: <judgement> "Category": "Strict/Plausible/No", "Relation/Intuition": "write the captured relation/intuition here" </judgement>.

Figure 9: Prompting template for judging the soundness levels of extracted 1-premise rules. Note that, since we use DeepSeek-R1 as our judge model, we put this instruction directly in the User role instead of the System part as suggested by the official document.

**Task**

For the given paired premises $P_1, P_2$ and conclusion $C$, judge whether the implication

$$P_1 \wedge P_2 \rightarrow C$$

is a **Strict or Plausible Horn Clause**, i.e., whether the co-occurrence of premises $P_1$ and $P_2$ in the front can point toward the occurrence of conclusion $C$ later in solving mathematical problems. You should classify the given horn clause candidate into one of "Strict/Plausible/No". Here, "Strict Horn Clauses" capture causally and logically relations (e.g., mathematical theorems), while "Plausible Horn Clauses" capture helpful intuitions or heuristic strategies to solve math problems (e.g., planning and checking) *without* strict logical soundness required. If the horn clause candidate does *not* reflects any strict relations or plausible intuitions, classify it as "No", indicating not a horn clause.

**First Premise ($P_1$)** **
[[ Insert Premise 1 Here ]]

**Second Premise ($P_2$)** **
[[ Insert Premise 2 Here ]]

**Conclusion ($C$)**

[[ Insert Premise 3 Here ]]

**Output**
Organize your final judgement as a **JSON** object with the following keywords:
"Category": select from "Strict/Plausible/No", "Relation/Intuition": a string less then 25 words
The JSON object should be wrapped by a special tag: <judgement> "Category": "Strict/Plausible/No", "Relation/Intuition": "write the captured relation/intuition here" </judgement>.

Figure 10: Prompting template for judging the soundness levels of extracted 2-premises rules. Note that, since we use DeepSeek-R1 as our judge model, we put this instruction directly in the User role instead of the System part as suggested by the official document.

