# OpenReview forum: "Soundness-Aware Level: A Microscopic Signature that Predicts LLM Reasoning Potential"
_ICLR.cc/2026/Conference — Submitted to ICLR 2026_

### Official Review · Reviewer_jKNw · 2025-10-30

**Soundness:** 4
**Presentation:** 4
**Contribution:** 4
**Rating:** 8
**Confidence:** 4

**Summary:**

This paper proposes Soundness-Aware Level (SAL), a metric designed to predict a large language model’s reasoning ability after reinforcement learning with verifiable rewards (RLVR). Experiments across multiple model families show that SAL strongly correlates with post-RLVR reasoning performance.

**Strengths:**

1. Logical rules (Horn clauses) are extracted from sparse representations of model activations, and their soundness is assessed based on whether each rule “makes sense” intuitively. The model’s reasoning potential is then evaluated by measuring how distinctly it separates sound from unsound logical rules in its internal feature space. It is an appealingly simple and intuitive idea.
2. The paper carefully explains each stage of the proposed pipeline — training sparse autoencoders on hidden activations, extracting co-activation rules, labeling their logical soundness, and computing the Soundness-Aware Level (SAL) via Jensen–Shannon divergence. Each step is well-motivated and supported with sufficient detail, making it easy to follow and pleasant to read.
3. The experiments align well with the authors’ hypothesis. The results consistently show that higher SAL values correlate strongly with better post-RLVR reasoning performance across multiple model families.

**Weaknesses:**

The approach relies on LLMs to label and categorize logical rules. This may introduce circularity or bias.

**Questions:**

Why do the authors use the term “Mechanism interpretation” rather than mechanistic interpretability or mechanistic interpretation?

---

> ### Author Response · Authors · 2025-11-14
> **Official Response to Reviewer jKNw**
>
> We thank the reviewer for recognizing the intuitive nature of our methodology, the clarity and thoroughness of the overall pipeline, and the strong empirical support across model families. We address the remaining points below.
>
> * __W1: The approach relies on LLMs to label and categorize logical rules. This may introduce circularity or bias.__
>
>   Ideally, assessing rule soundness requires human annotators. However, given the scale of extracted rules, human annotation is infeasible in practice. In our study, we therefore adopt DeepSeek-R1 as a machine annotator under the empirical assumption that its judgments closely approximate those of human annotators, which is partially supported by our human agreement analysis in Appendix D (Lines 850-855). Under this assumption, the procedure is not circular: DeepSeek-R1 acts as a scalable surrogate for human evaluation rather than as part of a model self-evaluation loop. In addition, we would highlight that applying modern LLMs to serve as a machine annotator for conducting large-scale interpretation analysis has become an important approach in mechanistic interpretability [1, 2, 3].
>
>   To make this assumption explicit and avoid confusion for further readers, we have added a clarification in the main text (Lines 188-190): "Ideally, these semantic judgments would be provided by human annotators, but the scale of extracted rules makes this infeasible. We therefore use a high-capability LLM as a scalable surrogate for human evaluation (see Appendix D for details)."
>
>   > [1] Steven Bills, et al. "Language models can explain neurons in language models." OpenAI Blog. 2023.
>   >
>   > [2] Lieberum, Tom, et al. "Gemma Scope: Open Sparse Autoencoders Everywhere All At Once on Gemma 2." Proceedings of the 7th BlackboxNLP Workshop: Analyzing and Interpreting Neural Networks for NLP. 2024.
>   >
>   > [3] Templeton, et al., "Scaling Monosemanticity: Extracting Interpretable Features from Claude 3 Sonnet", Transformer Circuits Thread, 2024.
>
> * __Q1: Why do the authors use the term “Mechanism interpretation” rather than mechanistic interpretability or mechanistic interpretation?__
>
>   Thanks for pointing out the typos. Our revision changes the phrases to mechanistic interpretation.

---

### Official Review · Reviewer_gLaR · 2025-10-31

**Soundness:** 2
**Presentation:** 3
**Contribution:** 2
**Rating:** 2
**Confidence:** 4

**Summary:**

This paper aims to find a score for selecting a suitable pretrained model for RLVR. They find that the soundness-aware level (SAL) is indicative of the post-RL performance, which roughly measures whether the model can tell if a logical implication is sound vs not.

The method takes the following steps:
- Extract features from hidden representations using SAE.
- Construct logical implication rules from the features by checking co-occurrences patterns, and use a LLM (DeepSeek-R1) to label each rule as Strict / Plausible / Noise.
- Compare how the distributions over Strict / Plausible / Noise rules are different, which is quantified by the JSD on histograms.

Intuitively, the more different these distributions are, the higher the score, suggesting the model is more "soundness-aware".

Empirically, SAL correlates well with post-RL performance on 7 models. 4 of them are from the Qwen family with different sizes, and 3 models are around 7B scale, from Mistral, Llama, and DeepSeek.

**Strengths:**

- This is one of the first papers that consider the problem of choosing base model for RL, which is timely.
- The experiments consider different model families and tasks.

**Weaknesses:**

I'm mainly concerned about limited empirical evidence.
- There are only 7 models, so the correlation number is not convincing.
- The trend is especially unclear with 7B-scale models across families, i.e. no clear ordering at the lower left corner of Fig 4 (left).
- The paper fits a power law curve between SAL and the post-RL performance, but the empirical data only supports the near-linear regime of the curve, so it's unclear why a power law is justified.
- There's no quantification of uncertainty/variation. For example, if we train different SAEs on the same model, how much difference will there be in SAL scores?

**Questions:**

- It seems that the bigger model is better based on Qwen results. Is this always true?
  - If yes, this (i.e. "bigger is better") is not providing new information unless SAL can quantify the amount of improvement -- though I'm doubtful about the accuracy of the power law; please see my comment in Weakness.
  - If no, please comment on when & how SAL can identify the proper size.
- Quantifying uncertainty/variations due to randomness: If we train different SAEs on the same model, how much difference will there be in SAL scores?
- Please report the computational cost of the method.

---

> ### Author Response · Authors · 2025-11-14
> **Official Response to Reviewer gLaR (Part 1/2)**
>
> We thank the reviewer for recognizing the timeliness of studying base-model selection for RLVR, as well as the value of evaluating our method across multiple model families. We address your raised concerns point by point below.
>
> * __W1: There are only 7 models, so the correlation number is not convincing.__
>
>   We kindly disagree with the argument that the correlation scores in Table 1 are not convincing. Although the number of publicly available RLVR-trained models is limited, the correlations are statistically robust: we constantly observe that the Spearman correlation is sufficiently strong to reject the null hypothesis that SAL and post-RL performance are independent (with p-value < 0.01) across all datasets. __This confirms that our extracted microscopic logic rules can reveal macroscopic reasoning behaviors.__ Our revision has included these significance indicators to reduce any potential concern from future readers in Line 343.
>
> * __W2: The trend is especially unclear with 7B-scale models across families, i.e. no clear ordering at the lower left corner of Fig 4 (left).__
>
>   We believe there is a misunderstanding in the left corner of Figure 4 (Left), where the three data points are not solely 7B models. The comparison of 7B models from different families is reported in Figure 5 (right). From which we can observe that different model families demonstrate significantly different SAL scores. This observation suggests that the training corpus, architectures, and other factors can also significantly impact the reasoning potential of base models.
>
> * __W3: The paper fits a power law curve between SAL and the post-RL performance, but the empirical data only supports the near-linear regime of the curve, so it's unclear why a power law is justified.__
>
>   When modeling this empirical correlation, we first consider the two theoretical bounds of SAL described in Lines 361 to 363, namely the case where rule distributions completely overlap and the case where they are perfectly separated. We then evaluate a range of functional forms, including linear, exponential, logistic, polynomial, and piecewise-linear relationships, and report their results in Table 1. Although polynomial and piecewise-linear functions achieve high in-sample R-squared, they fail to respect the oracle ceiling boundary implied by our metric, and their leave-one-out R-squared values become strongly negative, which indicates severe overfitting. In contrast, simpler monotonic functions from the exponential family fit both theoretical bounds and exhibit stable generalization performance. This motivates our choice of the exponential form for summarizing the observed empirical correlation. More discussions about choices of exponential power model can be found in Lines 363-370.
>
>
>   __Table 1: Leave-one-out (LOO) evaluation of different functional forms across ALL data points (7 observed and 2 theoretical).__
>   | Model                           | R² (Train) | R² (LOO)   |
>   | ------------------------------- | ---------- | ---------- |
>   | Exponential Power (Current Choice) | 0.9849   | 0.9715   |
>   | Logistic                        | 0.9796   | 0.9691   |
>   | Hill                            | 0.9778   | 0.9628   |
>   | Michaelis                       | 0.9550   | 0.9043   |
>   | Random                          | 0.000000   | -0.2657  |
>   | Linear                          | 0.7568   | -12.5192 |
>   | Polynomial                      | 0.9876   | -20.0756 |
>   | Piecewise linear | 0.9882   | -27.8406 |
>
> * __W4: There's no quantification of uncertainty/variation. For example, if we train different SAEs on the same model, how much difference will there be in SAL scores?__
>
>   We appreciate the reviewer’s point regarding the quantification of SAL's uncertainty. Quantifying this variance would indeed be informative, but training SAEs multiple times is computationally prohibitive.
>
>   More importantly, we would like to clarify the role of SAL in our work. **The primary goal of our experiments is to show that the rules extracted by our method capture structure that is closely related to the model’s reasoning behavior.** The SAL–performance correlations are therefore *evidence* that these extracted rules carry meaningful reasoning signals. SAL is one application of the extracted rules, not the main objective of our method itself.
>
>   Despite this, the SAL correlations are highly consistent across all models and datasets (p < 0.01), suggesting that the observed relationships are stable enough for the purpose they serve in this paper. A systematic study of robustness across repeated SAE trainings is valuable future work when computational resources permit.

---

> ### Author Response · Authors · 2025-11-14
> **Official Response to Reviewer gLaR (Part 2/2)**
>
> * __Q1: It seems that the bigger model is better based on Qwen results. Is this always true? If yes, this (i.e. "bigger is better") is not providing new information unless SAL can quantify the amount of improvement -- though I'm doubtful about the accuracy of the power law; please see my comment in Weakness.__
>
>   Yes, according to our results on the Qwen family reported in Figure 5 (Left), the bigger model is better for reasoning, while we can also observe that certain model families (e.g., Llama and Mistral) indicate a significantly low SAL in Figure 5 (Right). The main message we would like to deliver is: Simply scaling model scale can only **marginally improve** their reasoning capability further, considering other factors like architecture and pre-training data quality could be more effective way. The conclusion is draw from the observation from both Figure 5 (left) and (right) where we see that the improvement of SAL along with model size falls into a exponential relation, while same-scale models from different families show significant diverse SALs. This discussion has been stated in Lines 388-389 and 406-408.
>
> * __Q2: Quantifying uncertainty/variations due to randomness: If we train different SAEs on the same model, how much difference will there be in SAL scores?__
>
>   Please refer to the response for W4.
>
> * __Q3: Please report the computational cost of the method.__
>
>   Appendix E discusses the computing resources we used for our experiments. In particular, our experiments can be reproduced with the computing node with the following configuration. Each node is equipped with 96 virtual CPU cores, 1TB of main memory, 8 TB of cloud-connected disk space, and 8 A100 Nvidia GPUs with 80GB of GPU memory each. To train the cross-layer SAE for our largest candidate LLM (Qwen-2.5-14B), this single node requires training for around 60 hours. To extract the logic rules with the trained SAE, it takes about 50 hours and requires a maximum of 500 GB of main memory, while it does not require too many GPU resources. The time of extracting logic rules can be linearly reduced by increasing the number of computing nodes.

---

> > ### Comment · Reviewer_gLaR · 2025-11-26
> >
> > Thank you for the clarifications. I'm still concerned that 7 models are not sufficient evidence, though I understand the computational constraints and that there is probably not much the authors could do at this stage. I'd like to focus on two other main concerns instead:
> >
> > **About the main contribution**: I thought the main contribution is a score (SAL, which is the title) for choosing a good base model for RL (i.e. strong correlation to the post-RL performance), but the rebuttal says that "SAL is one application of the extracted rules, not the main objective of our method itself." Could you please clarify?
> >
> > **About the computation cost**: Thanks for pointing to Appendix E, which I missed before. The numbers confirm my concern that the proposed method is computationally expensive, e.g. 880 GPU hours for the results on Qwen-2.5-14B. It would be helpful if the paper could report the percentage of GPU hours required to compute SAL compared to the hours required for RLing the model.

---

> > > ### Author Response · Authors · 2025-11-27
> > > **Official Response to Follow-up Comments by Reviewer gLaR**
> > >
> > > Thanks for the follow-up comments.
> > >
> > > * __About the main contribution: I thought the main contribution is a score (SAL, which is the title) for choosing a good base model for RL (i.e. strong correlation to the post-RL performance), but the rebuttal says that "SAL is one application of the extracted rules, not the main objective of our method itself." Could you please clarify?__
> > >
> > >   Thanks for the opportunity to clarify our main contribution. The technical contributions of this research are three folds: (1) Formalizing the microscopic reasoning of LLMs as a chain of logic rules (Section 2.1), (2) Proposing empirically extract these logic rules with a cross-layer sparse autoencoder (Section 2.2-2.3), (3) Quantifying the reasoning potential of LLMs with extracted rules (Section 2.4), named as SAL. While the motivation of this research is to study the microscopic differences of LLMs to quantify their reasoning potential, we would highlight that __the foundations of computing SAL (Section 2.1-2.3) also provide a new perspective to understand the reasoning process from model internals__. These techniques for extracting logic rules can also benefit future research on understanding LLM reasoning behaviors.
> > >
> > > * __About the computation cost: Thanks for pointing to Appendix E, which I missed before. The numbers confirm my concern that the proposed method is computationally expensive, e.g. 880 GPU hours for the results on Qwen-2.5-14B. It would be helpful if the paper could report the percentage of GPU hours required to compute SAL compared to the hours required for RLing the model.__
> > >
> > >   Thanks for reviewing Appendix E. We would like to clarify a misunderstanding: the reported A100 GPU hours for Qwen-2.5-14B include both SAE training and logic-rule extraction, but only the SAE training (480 A100 hours) actually requires GPUs. The additional 50 hours for rule extraction can be executed on CPUs with nearly identical runtime, so they should not be counted as GPU cost.
> > >
> > >   For comparison, according to [1], GRPO training on Qwen-2.5-7B/14B with two 8×H100 nodes over 8K samples takes more than 15 hours, which is approximately equivalent to 400 A100 GPU hours [2]. This comparison suggests that computing SAL is a reasonably efficient way to estimate the reasoning potential of a pre-trained model, especially relative to the cost of full RL training. In addition, the trained SAE can support other analyses on the pre-trained models.
> > >
> > >   > [1] [hkust-nlp/simpleRL-reason: Simple RL training for reasoning](https://github.com/hkust-nlp/simpleRL-reason)
> > >   >
> > >   > [2] [H100 GPU | NVIDIA](https://www.nvidia.com/en-us/data-center/h100/)

---

### Official Review · Reviewer_9NuL · 2025-10-31

**Soundness:** 1
**Presentation:** 3
**Contribution:** 2
**Rating:** 2
**Confidence:** 4

**Summary:**

This is the updated review, incorporating a critical assessment of the statistical validity of the paper's central claims.SummaryThe paper investigates why different base LLMs exhibit varied reasoning performance after Reinforcement Learning with Verifiable Rewards (RLVR). The authors hypothesize that this potential is tied to a model's pre-trained ability to internally distinguish sound from unsound knowledge. They introduce a "microscopic signature," the Soundness-Aware Level (SAL), to quantify this.

The methodology for calculating SAL is complex. It involves: (1) training cross-layer Sparse Autoencoders (SAEs) to extract features; (2) identifying implicit logic rules (Horn clauses) by analyzing feature co-occurrence probabilities; (3) using an external LLM judge (DeepSeek-R1) to categorize the soundness of these rules (Strict, Plausible, Noise); and (4) calculating SAL as the Jensen-Shannon Divergence (JSD) between the probability distributions of these categories.The central claim is that SAL strongly predicts post-RLVR performance. Based on an analysis of **only** 7 models (4 Qwen, Llama, Mistral, DeepSeek), the authors claim the relationship between SAL and post-RLVR error rate follows a precise "empirical law" ($\epsilon=exp(-\alpha\cdot SAL^{\beta})$), reporting high R-squared values.

**Strengths:**

- The premise of the paper (understanding what makes a model RL-able) is an important problem. The proposed approach is new and interesting, using mechanistic interpretability.

- The framework, combining SAEs and logic, provides a novel and interesting method for analyzing the internal logic of LLMs. And the findings are

- SAL is proposed as an intrinsic signature that does not require labeled data from the downstream task (only an unlabeled corpus and the LLM judge), which makes it more usable than other model selection methods.

**Weaknesses:**

- The paper's assertion that the relationship between SAL and error rate follows a precise "empirical law" is a stretch! The law is derived from only 7 data points (N=7) with a heavily biased dataset of 4 out of the 7 models belong to the same Qwen family. It is too few points to claim anything about a law, in particular a piecewise linear function would fit this perfectly too.

- The SAL metric hinges entirely on an external LLM (DeepSeek-R1) categorizing the soundness of extracted rules. Appendix D reveals that the agreement between this LLM judge and human annotators is extremely low (0.566). If the foundational categorization of rules is unreliable, the SAL metric derived from these labels is inherently unstable and noisy.

- SAL is exceedingly complex and computationally intensive. It involves specialized SAE training, feature interpretation, and massive co-occurrence counting (Appendix D mentions $5.8\times10^{12}$ combinations). The computational resources make the approach infeasible.

**Questions:**

Address weaknesses above.

---

> ### Author Response · Authors · 2025-11-14
> **Official Response to Reviewer 9NuL (Part 1/2)**
>
> We thank the reviewer for highlighting the importance of the research problem, the novelty of our SAE-based approach to studying reasoning, and the clarity of our presentation. We address each of the raised concerns in detail below.
>
> * __W1.1: The paper's assertion that the relationship between SAL and error rate follows a precise "empirical law" is a stretch! The law is derived from only 7 data points (N=7) with a heavily biased dataset of 4 out of the 7 models belong to the same Qwen family.__
>
>   The observed strong correlation between SAL and post-RL error rate __indicates that our extracted microscopic logic rules can reveal macroscopic reasoning behaviors. It is too few points to claim anything about a law,__ To avoid any overstatement, our revision has replaced the term “empirical law’’ with “empirical correlation’’ throughout the manuscript.
>
>   Although the dataset contains only seven models, it is still informative for examining correlation trends. Prior empirical findings in the literature, such as early scaling results relating model performance to dataset size [1], were also initially supported by a similar number of data points. In addition, our seven models span four different model families, as noted in Lines 239-240, which is intended to evaluate the generalizability of SAL across architectures rather than to rely on a single lineage.
>
>   > [1] Kaplan, Jared, et al. "Scaling laws for neural language models." arXiv preprint arXiv:2001.08361 (2020).
>
> * __W1.2: ..., in particular a piecewise linear function would fit this perfectly too.__
>
>   When modeling this empirical correlation, we first consider the two theoretical bounds of SAL described in Lines 361 to 363, namely the case where rule distributions completely overlap and the case where they are perfectly separated. We then evaluate a range of functional forms, including linear, exponential, logistic, polynomial, and piecewise-linear relationships, and report their results in Table 1. Although polynomial and piecewise-linear functions achieve high in-sample R-squared, they fail to respect the oracle ceiling boundary implied by our metric, and their leave-one-out R-squared values become strongly negative, which indicates severe overfitting. In contrast, simpler monotonic functions from the exponential family fit both theoretical bounds and exhibit stable generalization performance. This motivates our choice of the exponential form for summarizing the observed empirical correlation. More discussions about choices of exponential power model can be found in Lines 363-370.
>
>   __Table 1: Leave-one-out (LOO) evaluation of different functional forms across ALL data points (7 observed and 2 theoretical).__
>   | Model                           | R² (Train) | R² (LOO)   |
>   | ------------------------------- | ---------- | ---------- |
>   | Exponential Power (Current Choice) | 0.9849   | 0.9715   |
>   | Logistic                        | 0.9796   | 0.9691   |
>   | Hill                            | 0.9778   | 0.9628   |
>   | Michaelis                       | 0.9550   | 0.9043   |
>   | Random                          | 0.000000   | -0.2657  |
>   | Linear                          | 0.7568   | -12.5192 |
>   | Polynomial                      | 0.9876   | -20.0756 |
>   | Piecewise linear | 0.9882   | -27.8406 |
>
> * __W2.1: The SAL metric hinges entirely on an external LLM (DeepSeek-R1) categorizing the soundness of extracted rules.__
>
>   Using DeepSeek-R1 to categorize the soundness of extracted rules is a practical decision driven by scale, as manually annotating tens of thousands of rules for the experiment is infeasible. Therefore, a strong LLM provides a consistent and scalable surrogate for constructing the soundness distributions required by SAL, and we already acknowledge this imperfect rather than a perfect solution in Appendix D. In addition, we would highlight that applying modern LLMs to serve as a machine annotator for conducting large-scale interpretation analysis has become a important approach in mechanistic interpretability [1, 2, 3]. Our revision has included these discussions in Lines 189-191 to avoid any confusion from broader audiences.
>
>   > [1] Steven Bills, et al. "Language models can explain neurons in language models." OpenAI Blog. 2023.
>   >
>   > [2] Lieberum, Tom, et al. "Gemma Scope: Open Sparse Autoencoders Everywhere All At Once on Gemma 2." Proceedings of the 7th BlackboxNLP Workshop: Analyzing and Interpreting Neural Networks for NLP. 2024.
>   >
>   > [3] Templeton, et al., "Scaling Monosemanticity: Extracting Interpretable Features from Claude 3 Sonnet", Transformer Circuits Thread, 2024.

---

> ### Author Response · Authors · 2025-11-14
> **Official Response to Reviewer 9NuL (Part 2/2)**
>
> * __W2.2: Appendix D reveals that the agreement between this LLM judge and human annotators is extremely low (0.566). If the foundational categorization of rules is unreliable, the SAL metric derived from these labels is inherently unstable and noisy.__
>
>   Regarding the agreement of 0.566 reported in Appendix D, our qualitative analysis shows that most disagreements occur between the softer boundaries across adjacent categories, such as "No'' with "Plausible'' and "Plausible'' with "Strict''. In contrast, both humans and DeepSeek-R1 rarely confuse between "Strict'' rules with "No'' ones. Moreover, the random-guess baseline for this three-way task is 0.33, so a score of 0.566 indicates that the LLM annotator provides some informative labels rather than noise. Because SAL is computed from the overall separation of the three soundness distributions rather than from individual labels, small shifts across adjacent boundaries have a limited effect on the final metric. In addition, in spite of these imperfections, we can still achieve good end predictive power, indicating the strong robustness of our proposed SAL metric. Our revision has included this discussion in Lines 850-856.
>
> * __W3: SAL is exceedingly complex and computationally intensive. It involves specialized SAE training, feature interpretation, and massive co-occurrence counting (Appendix D mentions combinations). The computational resources make the approach infeasible.__
>
>   We would first like to clarify that SAL is developed within the scope of mechanistic interpretability, and our goal is to understand how reasoning emerges inside LLMs by analyzing their internal representations. Methods in this research area typically require specialized models such as SAEs and large-scale feature analyses [1], and SAL follows this established paradigm. We acknowledge that SAL requires SAE training and large-scale co-occurrence counting, but the SAE component can be reused across projects, and our engineering optimizations (Appendix E) make the rule extraction feasible on moderate compute. Importantly, SAL aims to capture reasoning at a microscopic level, a task that is inherently complicated. Given the significance of understanding how reasoning emerges inside LLMs, we believe this computing cost is acceptable, and improving efficiency is a natural direction for future work.
>
>   > [1] Lindsey, et al., "On the Biology of a Large Language Model", Transformer Circuits, 2025.

---

> > ### Comment · Reviewer_9NuL · 2025-11-27
> > **Response to Authors**
> >
> > Thanks for your response. Here are my thoughts:
> >
> > - I appreciate that the authors removed the empirical law claim and replaced it with empirical correlation. I understand that from the 7 points, your law that you fit has good LOO performance. I'm not sure what piecewise linear function you used, cause naively using one could lead to high LOO error. Irrespective, I still think the 7 models (4 being from the same family) are too few for any meaningful claim. In particular, if you see the trend, it is established essentially by the 4 Qwen models and the other models have less influence. This makes it more relevant purely for Qwen.
> > - Using LLMs as a judge is a common approach, but it does tie your observations to dependencies on this LLM. It does help to know that the performance of the LLM judge is weak at the boundaries. It's not immediately clear to me why this does not affect downstream performance. So clarity on that would be helpful.
> > - Yes, for the goal of purely interpretability or understanding, I understand the compute costs being high can be justified. My concern was more that the usefulness of this for model selection is limited.
> >
> > While I like the overall idea, I believe the authors need to use more model families and balance out the number of models per family to improve soundness of their approach. And within this set of 7 models, there might be simpler measures that also correlate with the trend. Therefore I'll stick to my score.

---

> > > ### Author Response · Authors · 2025-11-27
> > > **Official Response to Follow-up Comments by Reviewer 9NuL**
> > >
> > > Thanks for the follow-up comments.
> > >
> > > We would like to highlight that the technical contributions of this research are three folds: (1) Formalizing the microscopic reasoning of LLMs as a chain of logic rules (Section 2.1), (2) Proposing empirically extract these logic rules with a cross-layer sparse autoencoder (Section 2.2-2.3), (3) Quantifying the reasoning potential of LLMs with extracted rules (Section 2.4), named as SAL. While the motivation of this research is to study the microscopic differences of LLMs to quantify their reasoning potential, __the foundations of computing SAL (Section 2.1-2.3) also provide a new perspective to understand the reasoning process from model internals__. These techniques for extracting logic rules can also benefit future research on understanding LLM reasoning behaviors.

---

### Official Review · Reviewer_gH88 · 2025-11-02

**Soundness:** 2
**Presentation:** 3
**Contribution:** 2
**Rating:** 2
**Confidence:** 4

**Summary:**

This paper presents a new method for probing the internal logic of a pre-trained LLM. The method consists of three steps. In step 1, a cross-layer sparse autoencoder produces semantically meaningful features from the LLM’s hidden activations. In step 2, the method extracts implicit logical rules by computing feature co-occurrences and estimating conditional probabilities for entailment in if-then rules. In step 3, an LLM judge evaluates the quality of the extracted rules, and the method computes the Jensen-Shannon (JS) divergence between three soundness distributions: noise, plausible, and strict. The higher the JS divergence, the more effective the pre-trained model is at distinguishing sound statements from unsound ones. The paper presents experiments showcasing the effectiveness of the method.

**Strengths:**

The method is interesting: use features from sparse autoencoders to extract logical rules with co-occurrent probabilities, then separate the probability distributions.

**Weaknesses:**

-The claim that their method "successfully predicts the downstream reasoning potential of pre-trained language models after RL training" (Line 472) is overstated. The paper addresses correlation, not causation (see Line 484). Also, there may be unknown confounding factors. The paper does not have intervention or ablation studies.

-The sparse autoencoders are not perfect. Their error should be propagated forward, but this is not done.

- The paper assumes that the features extracted from the sparse autoencoder are monosemantic, which is usually not the case.

- Using an LLM to judge the soundness of rules extracted from other LLMs seems circular.

**Questions:**

- Why did you choose Jensen-Shannon divergence instead of Wasserstein distance?

- Why did you not use human validation instead of an LLM as a judge?

- I assume you did not run controlled experiments with the same architecture on different pre-training corpora, or vice versa, because you did not have the compute resources. Is that correct?

- How does your method generalize to out-of-distribution reasoning?

- Why does soundness-awareness correlate with reasoning?

---

> ### Author Response · Authors · 2025-11-14
> **Official Response to Reviewer gH88 (Part 1/2)**
>
> We thank the reviewer for recognizing our methodology that studies the reasoning behaviors from a microscopic perspective. We address each of the raised concerns in detail below.
>
> * __W1: The claim that their method "successfully predicts the downstream reasoning potential of pre-trained language models after RL training" (Line 472) is overstated. The paper addresses correlation, not causation (see Line 484). Also, there may be unknown confounding factors. The paper does not have intervention or ablation studies.__
>
>   Thanks for pointing this out. We did not intend to imply any causal relationship, as we have clearly stated in Lines 481-483: "While our work establishes a strong predictive correlation, providing a direct causal link between SAL and reasoning potential is a crucial direction for future research."
>
>   To avoid potential misunderstanding for broader audiences, our revision has revised Line 470 to clarify this point as: "SAL provides a reliable predictive signal for the downstream reasoning potentials of pre-trained language models after RL training."
>
> * __W2: The sparse autoencoders are not perfect. Their error should be propagated forward, but this is not done.__
>
>   We agree that SAEs are not ideal for perfectly reconstructing hidden representations, and indeed, they do not propagate reconstruction errors back into the forward pass. However, our method is unaffected by this limitation because SAL does not rely on reconstruction at all. Once an SAE is trained, we only monitor the activation patterns of the learned features (i.e., whether a feature fires), rather than using the SAE decoder. Therefore, reconstruction error does not enter into the computation of SAL.
>
>   To avoid confusion, our revision has added a clarification in Lines 176-177 explicitly noting that SAL is computed solely from feature activations as "Note that the computation uses only feature activations and does not rely on perfect latent-space reconstruction, which is unattainable in practice."
>
> * __W3: The paper assumes that the features extracted from the sparse autoencoder are monosemantic, which is usually not the case.__
>
>   We agree that SAE features are not guaranteed to be monosemantic in practice. This is precisely why we restrict SAL computation to the subset of features that exhibit reliable monosemantic explanations in our study. As detailed in Appendix C.2 (Lines 796–800), we filter for features that activate at least 30 times across the corpus and observe a 96.27% overall explained rate. This ensures that SAL is computed only from features whose semantics can be consistently interpreted.
>
>   To make this assumption explicit to readers, the revision includes a clarification in the main text in Lines 253-254: "To ensure meaningful analysis, we further focus on the SAE features that demonstrate reliable monosemantic explanations, as detailed in Appendix C.2."
>
> * __W4: Using an LLM to judge the soundness of rules extracted from other LLMs seems circular.__
>
>   Ideally, assessing rule soundness requires human annotators. However, given the scale of extracted rules, human annotation is infeasible in practice. In our study, we therefore adopt DeepSeek-R1 as a machine annotator under the empirical assumption that its judgments closely approximate those of human annotators, which is partially supported by our human agreement analysis in Appendix D (Lines 850-855). Under this assumption, the procedure is not circular: DeepSeek-R1 acts as a scalable surrogate for human evaluation rather than as part of a model self-evaluation loop. In addition, we would highlight that applying modern LLMs to serve as a machine annotator for conducting large-scale interpretation analysis has become an important approach in mechanistic interpretability [1, 2, 3].
>
>
>   To make this assumption explicit and avoid confusion for further readers, we have added a clarification in the main text (Lines 188-190): "Ideally, these semantic judgments would be provided by human annotators, but the scale of extracted rules makes this infeasible. We therefore use a high-capability LLM as a scalable surrogate for human evaluation (see Appendix D for details)."
>
>
>   > [1] Steven Bills, et al. "Language models can explain neurons in language models." OpenAI Blog. 2023.
>   >
>   > [2] Lieberum, Tom, et al. "Gemma Scope: Open Sparse Autoencoders Everywhere All At Once on Gemma 2." Proceedings of the 7th BlackboxNLP Workshop: Analyzing and Interpreting Neural Networks for NLP. 2024.
>   >
>   > [3] Templeton, et al., "Scaling Monosemanticity: Extracting Interpretable Features from Claude 3 Sonnet", Transformer Circuits Thread, 2024.

---

> ### Author Response · Authors · 2025-11-14
> **Official Response to Reviewer gH88 (Part 2/2)**
>
> * __Q1: Why did you choose Jensen-Shannon divergence instead of Wasserstein distance?__
>
>   Wasserstein distance is also a valid option for comparing two probability distributions. We choose Jensen–Shannon divergence because SAL requires comparing three distributions simultaneously (i.e., Strict, Plausible, Noise), and JSD has a standard and well-defined multi-distribution formulation with a corresponding mixture distribution. Using Wasserstein in this setting would require additional design choices, such as how to aggregate multiple pairwise distances, which introduces arbitrariness into the metric. JSD therefore offers a more direct and principled measure for our purposes.
>
> * __Q2: Why did you not use human validation instead of an LLM as a judge?__
>
>   Please see the response to Weakness 4.
>
> * __Q3: I assume you did not run controlled experiments with the same architecture on different pre-training corpora, or vice versa, because you did not have the computing resources. Is that correct?__
>
>   Yes. As discussed in Appendix E, computing SAL requires substantial resources, so training controlled variants with the same architecture or corpus was beyond our current computational budget.
>
>   We also understand that the proposed controlled experiments on different pre-training corpora aim to further probe the causal factors underlying changes in SAL. We have clarified in the revision that such causal investigations are planned as future work (Lines 481-483).
>
> * __Q4: How does your method generalize to out-of-distribution reasoning?__
>
>   If “out-of-distribution reasoning’’ refers to evaluating a model’s reasoning ability in domains different from those used to train the SAE, then SAL could be partially generalizable. SAL captures a model’s ability to separate sound from unsound rules in its internal representations, and many reasoning patterns are shared across domains. However, domain-specific reasoning rules may not be fully covered if the SAE is trained on a narrow corpus. We have added a clarification noting this limitation in Lines 484-485.
>
> * __Q5: Why does soundness-awareness correlate with reasoning?__
>
>   We interpret our findings as follows. In RLVR methods such as GRPO, the model is rewarded only for producing the correct final answer, without any reward for intermediate reasoning steps. For the model to benefit from such feedback, its internal representations must differentiate reasoning patterns that tend to lead to correct answers (i.e., sound rules) from those that frequently cause errors (i.e., spurious rules). Otherwise, the final reward signal is applied uniformly to different rules and cannot consistently encourage the model to use the sound ones. A higher SAL indicates that the model has already learned to separate sound and spurious rules during pre-training, even though it may not yet know how to use these rules effectively for problem solving. RLVR then builds on this structural separation to reinforce the use of sound rules, leading to improved reasoning performance.

---

### Author Response · Authors · 2025-11-14
**Global Response to Reviewers**

We sincerely thank the reviewers for their detailed and insightful feedback. Below, we summarize the key strengths of our manuscript as noted by the reviewers:

- The research problem of choosing base models for RL training is important and timely. (Reviewer 9NuL, gLaR)
- The proposed method that utilizes features from SAEs to study logic reasoning is novel and intuitive. (Reviewer gH88, 9NuL, jKNw)
- The experiment considers the generalizability of the proposed method across different model families. (Reviewer gLaR, jKNw)
- The presentation of the manuscript is good. (Reviewer gH88, 9NuL, gLaR, jKNw)

We respond to each reviewer's concerns in their respective comment sections.

---

### Meta-Review · Area_Chair_Dnyx · 2026-01-07

**Summary:**

This paper introduces Soundness-Aware Level (SAL), a metric designed to predict LLM reasoning potential after RLVR by measuring how distinctly models separate sound from unsound logical rules in their internal representations. While the research question of understanding what makes base models suitable for RL training is timely and the approach of using sparse autoencoders to extract logical rules is novel, the paper suffers from fundamental limitations that undermine its claims. The empirical evidence rests on only 7 models, with 4 belonging to the same Qwen family, which is insufficient to establish the claimed correlation as a generalizable finding. The reliance on an LLM judge (DeepSeek-R1) with modest human agreement (0.566) raises concerns about the reliability of the soundness categorization. Additionally, the computational cost (approximately 480 GPU hours for SAE training on a 14B model) limits practical applicability as a model selection tool. These issues, combined with the lack of causal validation and potential overfitting concerns with more flexible functional forms, indicate that the paper's contributions require stronger empirical grounding before acceptance.

**Reviewer Concerns:**

The authors adequately addressed concerns regarding the clarification of correlation versus causation claims, the terminology adjustment from "empirical law" to "empirical correlation," and the explanation of why reconstruction error does not affect SAL computation. The clarifications about focusing on monosemantic features and the rationale for using Jensen-Shannon divergence were also satisfactory.

However, critical concerns remain unresolved. The limited sample size of 7 models with heavy Qwen family bias continues to be problematic. Reviewer 9NuL specifically noted that the trend is essentially established by Qwen models, limiting generalizability to other families. While authors explained that the 0.566 agreement rate is acceptable for distribution-level analysis since disagreements occur mainly at adjacent category boundaries, this justification requires further empirical validation. The computational expense, though comparable to RL training costs as authors noted, still poses practical concerns for iterative model selection. The absence of uncertainty quantification for SAL scores across different SAE trainings remains unaddressed due to computational constraints.

**Reviewer Scores:**

Reviewer gH88 (Score: 2): Would likely maintain their score. While clarifications were provided regarding correlation claims and SAE usage, their core concerns about limited intervention studies, potential circularity in LLM-based evaluation, and the absence of controlled experiments remain unaddressed.

Reviewer 9NuL (Score: 2): Explicitly maintained their score after discussion, stating that more model families with balanced representation are needed and that simpler measures might achieve similar correlation within the limited dataset.

Reviewer gLaR (Score: 2): Would likely maintain their score. Their follow-up comments indicate continued concern about the insufficient number of models, and they requested clarification on the main contribution, suggesting unresolved conceptual issues.

Reviewer jKNw (Score: 8): Would likely maintain their positive assessment, as their concerns about LLM-based labeling were addressed and they found the methodology intuitive and well-presented.

---

### Decision · Program_Chairs · 2026-01-26

Reject